# Multi-Parametric Cardiac Magnetic Resonance for Prediction of Heart Failure Death in Thalassemia Major

**DOI:** 10.3390/diagnostics13050890

**Published:** 2023-02-26

**Authors:** Antonella Meloni, Laura Pistoia, Maria Rita Gamberini, Liana Cuccia, Roberto Lisi, Valerio Cecinati, Paolo Ricchi, Calogera Gerardi, Gennaro Restaino, Riccardo Righi, Vincenzo Positano, Filippo Cademartiri

**Affiliations:** 1Department of Radiology, Fondazione G. Monasterio CNR-Regione Toscana, 56124 Pisa, Italy; 2Unità Operativa Complessa Bioingegneria, Fondazione G. Monasterio CNR-Regione Toscana, 56124 Pisa, Italy; 3Unità Operativa Semplice Dipartimentale Ricerca Clinica, Fondazione G. Monasterio CNR-Regione Toscana, 56124 Pisa, Italy; 4Dipartimento della Riproduzione e dell’Accrescimento Day Hospital della Talassemia e delle Emoglobinopatie, Azienda Ospedaliero-Universitaria Arcispedale “S. Anna”, 44124 Cona, Italy; 5Unità Operativa Complessa Ematologia con Talassemia, Azienda di Rilievo Nazionale ad Alta Specializzazione Civico “Benfratelli-Di Cristina”, 90134 Palermo, Italy; 6Unità Operativa Dipartimentale Talassemia, Presidio Ospedaliero Garibaldi-Centro—ARNAS Garibaldi, 95100 Catania, Italy; 7Struttura Semplice di Microcitemia, Ospedale “SS. Annunziata”, 74123 Taranto, Italy; 8Unità Operativa Semplice Dipartimentale Malattie Rare del Globulo Rosso, Azienda Ospedaliera di Rilievo Nazionale “A. Cardarelli”, 80131 Napoli, Italy; 9Unità Operativa Semplice di Talassemia, Presidio Ospedaliero “Giovanni Paolo II”—Distretto AG2 di Sciacca, 92019 Sciacca, Italy; 10Unità Operativa Complessa Radiodiagnostica, Gemelli Molise SpA—Fondazione di Ricerca e Cura “Giovanni Paolo II”, 86100 Campobasso, Italy; 11Diagnostica per Immagini e Radiologia Interventistica, Ospedale del Delta, 44023 Lagosanto, Italy

**Keywords:** thalassemia major, cardiac magnetic resonance, iron overload, heart failure, death, prognosis

## Abstract

We assessed the prognostic value of multiparametric cardiovascular magnetic resonance (CMR) in predicting death from heart failure (HF) in thalassemia major (TM). We considered 1398 white TM patients (30.8 ± 8.9 years, 725 women) without a history of HF at baseline CMR, which was performed within the Myocardial Iron Overload in Thalassemia (MIOT) network. Iron overload was quantified by using the T2* technique, and biventricular function was determined with cine images. Late gadolinium enhancement (LGE) images were acquired to detect replacement myocardial fibrosis. During a mean follow-up of 4.83 ± 2.05 years, 49.1% of the patients changed the chelation regimen at least once; these patients were more likely to have significant myocardial iron overload (MIO) than patients who maintained the same regimen. Twelve (1.0%) patients died from HF. Significant MIO, ventricular dysfunction, ventricular dilation, and replacement myocardial fibrosis were identified as significant univariate prognosticators. Based on the presence of the four CMR predictors of HF death, patients were divided into three subgroups. Patients having all four markers had a significantly higher risk of dying for HF than patients without markers (hazard ratio (HR) = 89.93; 95%CI = 5.62–1439.46; *p* = 0.001) or with one to three CMR markers (HR = 12.69; 95%CI = 1.60–100.36; *p* = 0.016). Our findings promote the exploitation of the multiparametric potential of CMR, including LGE, for better risk stratification for TM patients.

## 1. Introduction

Beta thalassemia major (β-TM) is a genetic blood disease with a high incidence in the Mediterranean basin, the Middle East, the Indian subcontinent, Central Asia, and the Far East [1]. However, due to the increased migration flux, thalassemia has become a global health problem. β-TM is characterized by a reduced or absent synthesis of the β chains of hemoglobin with a consequent excess of α chains which aggregate and precipitate in the red cells, leading to chronic hemolysis and the destruction of red cells and their precursors in the bone marrow or peripheral blood [2]. These abnormalities result in severe anemia, which needs lifelong regular blood transfusions. Due to the absence of a physiologic excretory pathway for excess iron, the major drawback of this treatment is iron overload, which, being highly cytotoxic, can cause organ dysfunction and damage [3]. Iron-induced heart failure (HF) remains the main cause of morbidity and mortality in TM patients, although the introduction of T2* Cardiovascular Magnetic Resonance (CMR) for the non-invasive assessment of myocardial iron overload (MIO) led to a significant increase in the survival rate [4,5]. Indeed, this technique offers the possibility to design tailor-made iron chelation therapies customized for each patient and to evaluate their efficacy [6,7,8,9]. In addition to direct myocardial injury, iron overload may affect the heart indirectly because hepatic dysfunction and endocrinopathies (diabetes mellitus, hypothyroidism, and hypoparathyroidism) arising from iron accumulation increase the risk for heart failure independently of cardiac iron status [10,11,12]. Nevertheless, the pathophysiology of heart failure in TM can be multifactorial with significant contributions from physiologic, immunoinflammatory, and genetic factors [13,14,15].

Thanks to its multiparametric potential, CMR represents a unique tool for the characterization and quantification of myocardial involvement and damage. CMR is the gold standard for the quantification of biventricular size and function by cine images. Because it does not incorporate ionizing radiation, does not exhibit window or geometric limitations, and provides precise ventricular endocardial definition, it allows for highly reproducible and accurate measurements of ventricular volumes. This is of particular value in TM, where the “normal” heart pumps at larger volumes and against lower peripheral resistances than the normal heart in non-thalassemic individuals, and where the heart’s biventricular size can be influenced by the iron previously accumulated. Moreover, late gadolinium enhancement (LGE) CMR is the only non-invasive imaging method that can detect replacement myocardial fibrosis, a common finding among TM patients [14,16,17].

A study of 481 Italian TM patients showed that heart iron, ventricular dysfunction, and replacement myocardial fibrosis could predict the future development of heart failure. Moreover, all three of these CMR markers remained independent prognosticators in a multivariate model that included a previous history of heart failure [11]. To the best of our knowledge, the association between CMR findings and HF death in TM patients has not yet been demonstrated.

The aim of this multicenter study was to evaluate the prognostic value of multiparametric CMR (cardiac iron, function, and replacement fibrosis) in predicting death from heart failure in a large cohort of well-treated TM patients.

## 2. Materials and Methods

### 2.1. Study Population

We considered 1485 TM patients (31.04 ± 8.88 years; 771 women) consecutively enrolled in the Myocardial Iron Overload in Thalassemia (MIOT) network, comprising 70 thalassemia centers and 10 magnetic resonance imaging (MRI) centers, where MRI exams were performed using homogeneous, standardized, and validated procedures [5,18]. The inclusion criteria of the MIOT project were: (1) male and female patients, of all ages, with thalassemia syndromes or structural hemoglobin variants, requiring MRI to quantify the cardiac and liver iron burden; (2) written informed consent; (3) written authorization for use and disclosure of protected health information; (4) no absolute contraindications to MRI.

All patients were from an Italian background and were uniformly treated. They had been regularly transfused since early childhood and started undergoing chelation therapy from the mid-to-late 1970s, whereas patients born after the 1970s received chelation therapy from early childhood.

All patients performed their first MRI scan between April 2006 and November 2015. All scans were performed in the week immediately prior to the scheduled blood transfusion. The clinical-anamnestic history of the patients, from birth to the date of the first MRI scan, was recorded in the MIOT web-based database. At every MRI follow-up, which was performed by protocol every 18 ± 3 months, the clinical, instrumental, and laboratory data were updated.

All patients gave informed consent in compliance with the Declaration of Helsinki, and the study was approved by the institutional ethics committees of all MRI sites.

### 2.2. Magnetic Resonance Imaging

All patients underwent MRI using the clinical 1.5 T scanners of three main vendors (GE Healthcare, Milwaukee, WI; Philips, Best, Netherlands; Siemens Healthineers, Erlangen, Germany) equipped with phased-array coils. Breath-holding in end-expiration and ECG-gating were used.

For iron overload assessment, a validated T2* gradient–echo multiecho sequence was used. The intersite, interstudy, intraobserver, and interobserver variability of the proposed methodology had been previously assessed [19,20]. For the measurement of MIO, a multislice approach was adopted [21]. Three parallel short-axis views (basal, medium, and apical) of the left ventricle (LV) were acquired at 10 echo times (TE) (first TE = 2.0 ms, echo spacing = 2.26 ms) in a single end-expiratory breath-hold. Acquisition sequence details are provided in [22]. A medium hepatic slice was obtained at 10 TEs (echo spacing = 2.26 ms) in a single end-expiratory breath-hold [23]. T2* image analysis was performed by trained MRI operators (>10 years of experience) using a custom-written, previously validated software (HIPPO MIOT^®^) [24]. The software provided the T2* value for all the 16 segments of the LV, according to the standard American Heart Association (AHA)/American College of Cardiology (ACC) model [20]. The image analysis procedure included the manual delineation of the endocardial and epicardial borders of the LV wall, the identification of the upper intersection of the left and the right wall, and the automatic fitting of the signal decay over the TEs with an appropriate decay model. Susceptibility and geometric artifacts were corrected using an appropriate correction map [24]. The global heart T2* value was obtained by averaging all segmental values. Hepatic T2* values were calculated in a circular region of interest, defined in a homogeneous area of parenchyma without blood vessels [23], and were converted into liver iron concentration (LIC) with an appropriate calibration curve [25].

Steady-state free precession (SSFP) cines were acquired in sequential 8-mm short axis slices (gap 0 mm) from the atrio-ventricular ring to the apex to assess biventricular function parameters quantitatively in a standard way [26]. Thirty cardiac phases were acquired per heartbeat, and 10–14 slices were required to cover the heart over its entire extension. The most apical slice included was the first slice which showed no blood pool at end-diastole. The most basal slice included was the one that showed a remaining part of the thick myocardium and was below the aortic valve. The analysis was based on the manual recognition of the endocardial and epicardial borders of the wall, at least in the end-diastolic and end-systolic phases in each slice. Moreover, the papillary muscles were delineated and were considered myocardial mass rather than part of the blood pool. Biventricular volumes were indexed to the body surface area. The inter-center variability for the quantification of cardiac function was previously reported [27]. The left and right atrial areas were measured from the 4-chamber view projection in the ventricular end-systolic phase.

Late gadolinium enhancement short-axis images were acquired 10–18 min after Gadobutrol (Gadovist^®^; Bayer Schering Pharma; Berlin, Germany) intravenous administration at the standard dose of 0.2 mmol/kg using a fast gradient-echo inversion recovery sequence. In addition, vertical, horizontal, and oblique long-axis views were acquired. Inversion times were adjusted to null the normal myocardium (from 210 ms to 300 ms). LGE was evaluated visually by two independent observers using a two-point scale (enhancement absent or present) and was considered present when visualized in two different views [14]. LGE images were not acquired in patients with a glomerular filtration rate < 30 mL/min/1.73 m^2^ and in patients who refused the contrast medium administration.

### 2.3. Diagnostic Criteria and Follow-Up

A T2* measurement of 20 ms was taken as a “conservative” normal value for the segmental and global T2* values [28]. A LIC < 3 mg/g dry weight (dw) indicated no significant hepatic iron overload [29].

The mean serum ferritin level in the year preceding the MRI was taken into account, and a value ≥ 1000 ng/mL was considered indicative of significant body iron burden [30].

Previously derived reference ranges for biventricular volumes and function, specific to TM patients, were used [26]. Ventricular dilation was diagnosed in the presence of an LV and/or right ventricular (RV) end-diastolic volume index (EDVI) >2 standard deviations (SD) from the mean values normalized to age and gender. Ventricular dysfunction was diagnosed in the presence of an LV and/or RV ejection fraction (EF) <1 SD from the mean values normalized to age and gender.

Atrial dilatation was diagnosed if the left and/or right atrial area indexed by body surface area was ≥15 cm^2^/m^2^ [31].

The endpoint used in this study was HF-mortality. HF was identified based on symptoms (breathlessness, ankle swelling, and fatigue), signs, biomarkers, and instrumental parameters, according to the current guidelines [32].

The follow-up date coincided with the date of the last available MRI. For patients who did not perform a follow-up MRI, a case report form detailing patient outcomes between the baseline MRI and September 2018 was completed by the caring hematologist.

### 2.4. Statistical Analysis

All data were analyzed using SPSS version 27.0 (IBM Corp, Armonk, NY, USA) statistical package.

Continuous variables were described as mean ± SD. Categorical variables were expressed as frequencies and percentages.

The normality of the distribution of the continuous variables was assessed by using the Kolmogorov–Smirnov test.

For continuous values with normal distributions, comparisons between groups were made by performing the independent-samples t-test (2 groups) or a one-way analysis of variance (ANOVA) (>2 groups). Wilcoxon’s signed rank test or the Kruskal–Wallis test were applied for continuous values with non-normal distribution. χ^2^ testing was performed for non-continuous variables. The Bonferroni post hoc test was used for multiple comparisons between pairs of groups.

Correlation analysis was performed using Pearson’s test or Spearman’s test where appropriate.

The Cox proportional hazard model was used to test the association between the considered prognostic variables and the outcome (HF death). The results are presented as hazard ratios (HR) with 95% confidence intervals (CI). Kaplan–Meier curves were generated by relating the development of an outcome over time to each significant prognosticator. The log rank test was used to compare different strata in Kaplan–Meier analyses.

In all tests, a 2-tailed *p* < 0.05 was considered statistically significant.

## 3. Results

### 3.1. Selection of the Patients

At the baseline MRI, eighty-seven (5.9%) patients had a history of heart failure and were excluded from this study.

Compared to HF-free patients, patients with a history of HF were characterized at the baseline MRI by a significantly higher age (34.33 ± 7.69 years vs. 30.84 ± 8.91 years; *p* = 0.001), significantly lower global heart T2* values (23.82 ± 13.38 ms vs. 29.46 ± 12.03 ms; *p* < 0.0001), a significantly higher number of segments with a T2* < 20 ms (7.33 ± 7.04 vs. 4.47 ± 6.09; *p* < 0.0001), significantly higher LV EDVI (94.37 ± 23.97 mL/m^2^ vs. 86.60 ± 18.71 mL/m^2^; *p* = 0.005) and RV EDVI (93.15 ± 37.33 mL/m^2^ vs. 82.59 ± 18.73 mL/m^2^; *p* = 0.011), and significantly lower LV EF (57.72 ± 10.41% vs. 61.58 ± 7.09%; *p* < 0.0001) and RV EF (56.04 ± 10.17% vs. 61.37 ± 8.14%; *p* < 0.0001).

### 3.2. Patients’ Characteristics

Table 1 shows the demographic, clinical, and MRI features of the considered 1398 TM patients at the baseline MRI. The mean age was 30.8 ± 8.9 years, and 725 (51.9%) patients were women.

Bi-atrial areas were present for 1138 patients due to technical reasons.

The contrast medium was not administrated in 286 (20.5%) patients. Among the 187 (16.8%) patients with replacement myocardial fibrosis, none had an ischemic pattern, and two or more foci were detected in 59.9% of cases. The septum was involved in 80.6% of the cases. Patients with replacement myocardial fibrosis were significantly older than patients without replacement myocardial fibrosis (33.25 ± 7.79 years vs. 30.89 ± 8.48 years; *p* < 0.0001), but they showed comparable global heart T2* values (27.56 ± 12.62 ms vs. 29.06 ± 11.94 ms; *p* = 0.124).

At baseline, serum ferritin levels showed a significant positive correlation with MRI LIC values (R = 0.713; *p* < 0.0001) and a significant inverse correlation with global heart T2* values (R = −0.326; *p* < 0.0001). A significant inverse correlation was detected between global heart T2* and MRI LIC values (R = −0.303; *p* < 0.0001). Global heart T2* values were not correlated with biventricular volume indexes or LV cardiac indexes but showed a weak positive association with both LV EF (R = 0.182; *p* < 0.0001) and RV EF (R = 0.102; *p* = 0.005).

The mean follow-up time was 4.83 ± 2.05 years (median: 5.01 years).

### 3.3. Chelation Therapy

At the baseline MRI, patients received the following chelation regimens: deferoxamine (33.3%), deferiprone (19.0%), deferasirox (25.0%), combined deferoxamine + deferiprone (17.2%), sequential deferoxamine/deferiprone (5.0%), and others (0.5%).

During the follow-up, 49.1% of the patients changed their chelation regimen at least once, i.e., they switched to a different type of chelator or underwent modification of dose and/or frequency. Compared to patients who maintained the same regimen, those who changed the chelation regimen were more likely to have a baseline global heart T2* value < 20 ms (33.2% vs. 19.7%; *p* < 0.0001) and to have a baseline LIC ≥ 3 mg/g/dw (69.3% vs. 57.4%; *p* < 0.0001).

The percentage of patients with good compliance (correspondence between the time history of drug administration and the prescribed regimen > 60%) was significantly higher at the end of the study than at the baseline MRI (94.6% vs. 92.5%; *p* < 0.0001).

### 3.4. Patient Outcomes

Twelve (1.0%) patients died from heart failure. Ten patients had HF with reduced EF at echocardiography. The majority of them presented to the healthcare provider with a reduction in their effort tolerance due to dyspnea and/or fatigue. One patient presented not only with fatigue but also with chest pain and tachycardia and had elevated troponin levels. Two patients presented with palpitations. Two patients had chronic heart failures diagnosed >1 year after the CMR scan, that rapidly worsened. One patient had an HF with mildly reduced EF. One patient had HF with preserved EF and had evidence of structural heart disease. The mean age at death was 35.06 ± 8.68 years (range: 17–47 years).

Mean time from the baseline MRI to the HF-related death was 1.68 ± 1.78 years and, six (50.0%) deaths occurred within the first year of follow-up.

When compared to the other patients, patients who died by HF showed at the baseline MRI significantly higher serum ferritin levels and MRI LIC values, significantly lower global heart T2* values, a significantly higher numbers of segments with T2* < 20 ms, significantly lower biventricular EFs, and a significantly higher incidence of replacement myocardial fibrosis (Table 1).

One patient who died from HF had a previous history of myocarditis.

### 3.5. Prediction of Heart Failure Mortality

Table 2 shows the results of the univariate Cox regression analysis. No association was detected between age or gender and HF mortality. Significant MIO (global heart T2* < 20 ms), ventricular dysfunction, ventricular dilation, and replacement myocardial fibrosis were identified as significant univariate prognosticators. Figure 1 shows the Kaplan–Meier survival curves. The log-rank test revealed a significant difference in the curves for each prognosticator (significant MIO: *p* = 0.010, ventricular dysfunction: *p* = 0.030, ventricular dilation: *p* < 0.0001, and replacement myocardial fibrosis: *p* = 0.010).

Due to the low number of deaths for HF, it was not possible to perform a multivariate model. However, based on the presence of the four CMR prognosticators of HF death, patients were divided into three subgroups:(1)patients with none of the four CMR markers (group 0; *N* = 488);(2)patients with one to three CMR markers (group 1; *N* = 617);(3)patients with four CMR markers (group 2; *N* = 7).

Table 3 shows the comparison of the baseline data among the three groups. No difference in terms of age, age at the start of regular transfusions or chelation was detected. All patients with four CMR markers were male, whereas distribution by sex was homogeneous in the other two groups. Serum ferritin levels and MRI LIC values were significantly higher in group 1 than in group 0 (*p* < 0.0001 in both comparisons). Global heart T2* values were significantly lower in group 2 than in groups 0 and 1 (*p* < 0.0001 and *p* = 0.006, respectively) and in group 1 than in group 0 (*p* < 0.0001), whereas the number of segments with a T2* < 20 ms was significantly higher in group 2 than in groups 0 and 1 (*p* < 0.0001 and *p* = 0.018, respectively) and in group 1 than in group 0 (*p* < 0.0001). Significantly lower LV EF and RV EF values were found in group 2 compared to both group 1 (*p* < 0.0001 for both ventricles) and group 0 (*p* < 0.0001 for both ventricles) and in group 1 compared to group 0 (*p* < 0.0001 for both ventricles). LV EDVI and RVEDVI were significantly increased in group 2 compared to group 1 (*p* < 0.0001 for both ventricles) and to group 0 (*p* < 0.0001 for both ventricles) and in group 1 compared to group 0 (*p* < 0.0001 and *p* = 0.003, respectively). The frequency of replacement myocardial fibrosis was significantly higher in group 2 than in groups 1 and 0 and in group 1 than in group 0 (*p* < 0.0001 for all comparisons).

The frequency of HF death was significantly higher in group 2 than in both group 0 (14.3% vs. 0.2%; *p* < 0.0001) and group 1 (14.3% vs. 1.5%; *p* = 0.021) (Figure 2).

Patients having all four markers had a significantly higher risk of dying by HF than patients without markers (HR = 89.93; 95% CI = 5.62–1439.46; *p* = 0.001) or with one to three CMR markers (HR = 12.69; 95% CI = 1.60–100.36; *p* = 0.016). Figure 3 shows the Kaplan–Meier survival curve. The log-rank test revealed a significant difference in the curves (*p* < 0.0001).

## 4. Discussion

In our homogeneous white Italian/Mediterranean population, which had been well-treated since early childhood and followed for a mean of 4.8 years after the baseline MRI, we detected a low incidence of deaths from heart failure because the T2* report guided the patient-specific adjustment of the chelation regimen. Indeed, the patients who changed their chelation regimen (drug or frequency/dosage) during the follow-up were more likely to have significant MIO at baseline. Moreover, all MRI scans were performed after 2006, the year when a new era of chelation treatment had started thanks to the availability in the clinical arena of three different iron chelators and the evidence that they could be used in association to intensify chelation or make it more tolerable [33].

No prospective association was detected between hepatic iron or serum ferritin levels and HF mortality. These parameters cannot be used to infer cardiac iron status, as demonstrated by weak cross-sectional correlation with the cardiac T2* found in the present study and in other published studies [28,34,35,36]. The relationship between cardiac and hepatic iron is complex due to the differences in iron uptake and elimination between the two organs as well as the strong influence of both the type and pattern of chelation [6,9,37]. Cardiac T2* can identify preclinical cardiac iron deposition in patients with excellent control of total body iron stores [35,38,39,40].

As expected, MIO was a significant prognosticator of HF death. Excess iron can be detrimental to human cells through the production of hydroxyl radicals via Haber–Weiss–Fenton reactions, which cause oxidative damage to cellular components like lipids, proteins, and DNA [41,42]. Free iron can directly interact and interfere with a variety of ion channels of cardiomyocytes, including the L-type calcium channel, the ryanodine-sensitive calcium channel, voltage-gated sodium channel, and delayed rectifier potassium channel, making cardiomyocytes particularly vulnerable to iron overload. Excessive production of reactive oxygen species can also directly induce ferroptosis (a non-apoptotic mode of cell death) in cardiomyocytes by catalyzing the oxidation of phospholipids in the cell membrane [43]. Importantly, other CMR parameters, namely ventricular dilatation, ventricular dysfunction, and replacement myocardial fibrosis, also emerged as unfavorable prognosis determinants. Our findings are in line with the study by Pepe et al., where, in a multivariate model, replacement myocardial fibrosis, MIO, and ventricular dysfunction independently predicted non-fatal HF [11]. Initially, MIO may cause a reduction of ventricular dimensions through vascular and ventricular stiffening [44] but the ventricular systolic function can remain well preserved so that at the onset of the disease patients are generally asymptomatic. In end-stage disease, MIO may increase ventricular dimensions and decrease systolic function [45]. In the Italian TM population, replacement myocardial fibrosis was demonstrated to be a relatively common finding (~20%) [14,46,47], correlated with aging, negative cardiac remodeling, hepatitis C virus (HCV) infection, and diabetes mellitus in adult TM patients [10,14], and with lower cardiac T2* values in pediatric patients free of complications [40]. Moreover, a recent study showed an association between replacement fibrosis and decreased native T1 values measured by CMR [26], suggesting a potential pathophysiological role of MIO in the development of myocardial fibrosis. Indeed, native T1 mapping seems to have a higher sensitivity for low amounts of iron in comparison to the T2* technique. Although iron could be removed via chelation treatment, the induced heart damage may be not totally reversible. The findings of the present study further highlight the prognostic implications of replacement myocardial fibrosis. In fact, in different pathologies, such as dilatative cardiomyopathy, hypertrophic cardiomyopathy, aortic stenosis, and infiltrative diseases, replacement myocardial fibrosis represents a final common pathway of myocardial disease and is independently associated with cardiac and all-cause mortality [48].

Importantly, when the four CMR indices (cardiac iron, dilatation, dysfunction, and replacement fibrosis) were evaluated in combination, they fine-tuned the prognostic stratification of TM patients. Thus, the results of our study strengthen the usefulness of a multiparametric CMR approach which integrates biventricular ejection fractions and volumes and LGE with cardiac T2* to further ameliorate the prognosis of TM patients via the early identification of high-risk patients. Conversely, relying only on cardiac T2* as a unique marker of cardiac death may lead to suboptimal prognostic stratification.

It deserves mention that in our study, both ventricular dysfunction and dilation were diagnosed using previously derived “normal for TM” reference ranges in order to avoid a misdiagnosis of cardiomyopathy (underdiagnosis of dysfunction and overdiagnosis of dilatation) [26]. Indeed, despite transfusion therapy, TM represents a chronically anemic condition characterized by an elevation of blood volume (increased preload) and a decrease in systemic vascular resistance (decreased afterload) [49]. Both conditions enhance ventricular pump performance, and the anatomical–functional expression of this hemodynamic state is the enlargement of cardiac cavities [50].

### Limitations

This study suffers from several limitations.

The small number of HF deaths that occurred during the follow-up did not allow us to perform a multivariate analysis that included all variables identified in the univariate analysis. For this reason, we performed a model with the four CMR univariate prognosticators.

The prognostic value of the CMR mapping techniques (T1, T2, and extracellular volume) was not evaluated because they were not available at the time of patient enrolment.

We did not measure myocardial deformation (strain), which could be a more sensitive marker of myocardial dysfunction than EF [51]. Although feature-tracking (FT) CMR allows quantification of myocardial deformation on routine SSFP cine images, the dedicated post-processing FT software packages were not available in the MIOT centers.

More studies are needed to evaluate the transferability of our results to other TM populations with a lower prevalence of HCV infection, in which a lower frequency of myocardial fibrosis may be expected.

## 5. Conclusions

In TM patients, significant MIO, ventricular dysfunction, ventricular dilation, and replacement myocardial fibrosis were associated with a significantly higher risk of heart failure death, and the combined use of all four CMR indexes provided incremental prognostic information. Hence, the present study’s findings promote exploiting the multiparametric potential of CMR, including LGE, for better risk stratification for TM patients. Further studies are needed to verify if, in addition to the adjustment of iron chelation therapy, the adoption of treatment directed to myocardial performance may further open the prognosis of TM patients.

## Figures and Tables

**Figure 1 diagnostics-13-00890-f001:**
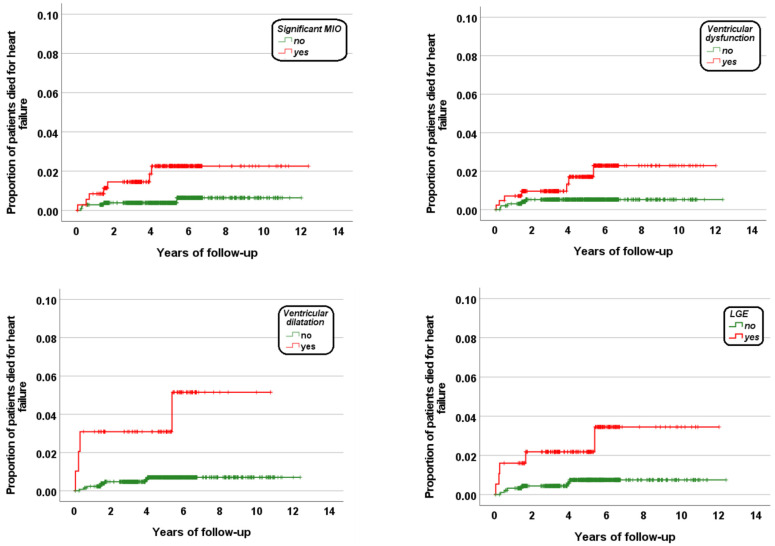
Kaplan–Meier curves showing the impact of each univariate predictive factor for HF death.

**Figure 2 diagnostics-13-00890-f002:**
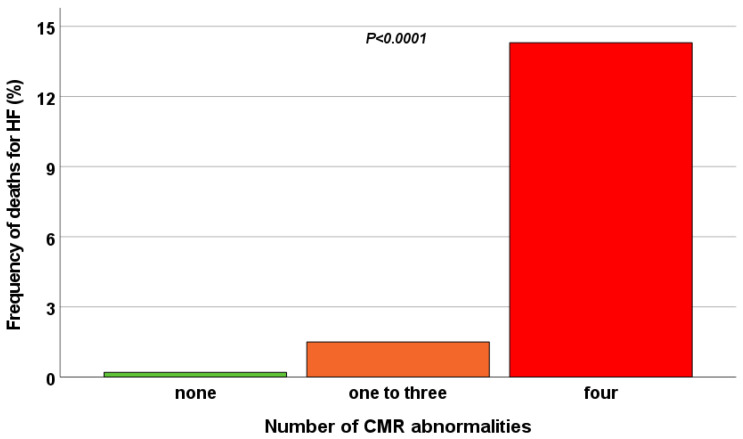
Frequency of HF death in the three groups identified based on the number of CMR abnormalities.

**Figure 3 diagnostics-13-00890-f003:**
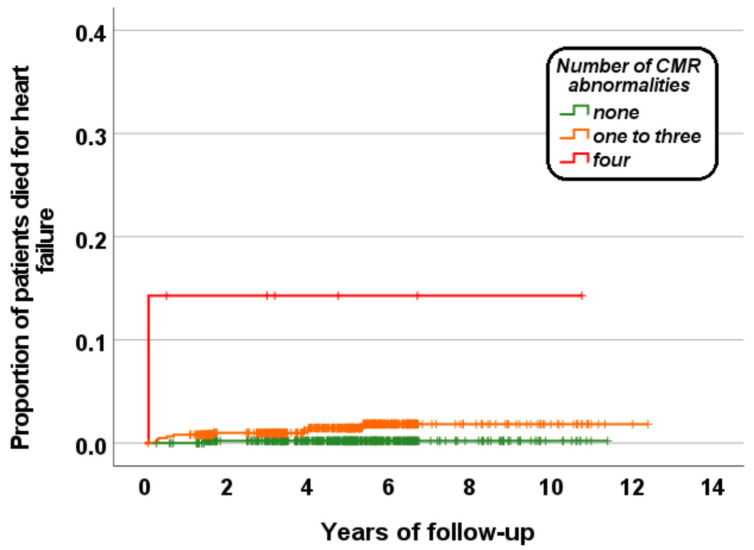
Kaplan–Meier curve for HF mortality as function of the number of CMR abnormalities (cardiac iron, dilatation, dysfunction, and replacement fibrosis).

**Table 1 diagnostics-13-00890-t001:** Demographic, clinical and MRI data at the baseline MRI.

Variable	All Patients	No HF Death	HF Death	*p*-Value
Women, *N* (%)	725 (51.9%)	719 (51.9%)	6 (50.0%)	0.897
Age (years)	30.84 ± 8.91	30.82 ± 8.92	33.38 ± 7.96	0.276
Age at start of regular transfusions (years)	2.24 ± 3.77	2.25 ± 3.78	1.40 ± 1.26	0.388
Chelation starting age (years)	4.69 ± 4.08	4.69 ± 4.08	4.67 ± 4.63	0.716
Pre-transfusion hemoglobin (g/dL)	9.58 ± 0.68	9.59 ± 0.68	9.13 ± 1.14	0.328
Serum ferritin (ng/mL)	1506.01 ± 476.65	1491.66 ± 1447.64	4589.17 ± 3607.63	0.012
Global heart T2* (ms)	29.46 ± 12.03	29.57 ± 11.97	17.50 ± 12.91	0.002
Number of segments with T2* < 20 ms	4.47 ± 6.09	4.43 ± 6.07	10.08 ± 6.30	0.002
MR LIC (mg/g/dw)	8.89 ± 10.87	8.81 ± 10.76	19.96 ± 16.98	0.009
LV EF (%)	61.58 ± 7.09	61.67 ± 6.89	51.41 ± 16.57	0.020
RV EF (%)	61.37 ± 8.14	61.47 ± 7.95	49.47 ± 16.98	0.009
LV EDVI (mL/m^2^)	86.60 ± 18.71	86.53 ± 18.53	95.03 ± 33.85	0.696
RV EDVI (mL/m^2^)	82.59 ± 18.73	82.51 ± 18.40	91.16 ± 29.49	0.414
LV cardiac index (L/min/m^2^)	3.68 ± 1.07	3.69 ± 1.07	3.53 ± 0.85	0.369
Replacement myocardial fibrosis, *N* (%)	187/1112 (16.8)	182/1101 (16.5)	5/11 (45.5)	0.025
Left atrial area (cm^2^/m^2^)	12.94 ± 2.65	12.93 ± 2.63	14.90 ± 4.06	0.139
Right atrial area (cm^2^/m^2^)	12.02 ± 2.35	12.02 ± 2.35	12.67 ± 1.92	0.264

*N* = number, MRI = magnetic resonance imaging, LIC = liver iron concentration, LV = left ventricular, EF = ejection fraction; RV = right ventricular, EDVI = end-diastolic volume index.

**Table 2 diagnostics-13-00890-t002:** Results of univariate Cox analysis for heart failure mortality.

	*N* (%) in Group	*N* (%) with HF	Univariate Analysis	
			HR (95%CI)	*p*
Sex				
Male	673 (48.1)	6 (0.9)	Reference	
Female	725 (51.9)	6 (0.8)	0.93 (0.29–2.87)	0.893
Age				
<31 years	645 (46.1)	4 (0.6)	Reference	
≥31 years	753 (53.9)	8 (1.1)	1.73 (0.52–5.75)	0.371
Serum ferritin				
<1000 ng/mL	667 (47.3)	3 (0.4)	Reference	
≥1000 ng/mL	731 (52.3)	9 (1.2)	4.58 (0.54–39.17)	0.165
MRI LIC				
<3 mg/g dw	512 (36.6)	2 (0.4)	Reference	
≥3 mg/g dw	886 (63.4)	10 (1.1)	2.89 (0.63–13.19)	0.171
Global heart T2*				
≥20 ms	1042 (74.5)	5 (0.5)	Reference	
<20 ms	356 (9.7)	7 (2.0)	4.04 (1.28–12.75)	0.017
Ventricular dysfunction				
no	978 (70.0)	5 (0.5)	Reference	
yes	420 (30.0)	7 (1.7)	3.30 (1.05–10.41)	0.041
Ventricular dilatation				
no	1301 (93.1)	8 (0.6)	Reference	
yes	97 (6.9)	4 (4.1)	6.63 (1.99–22.04)	0.002
Myocardial fibrosis (*N* = 1112)				
no	925 (83.2)	6 (0.6)	Reference	
yes	187 (16.8)	5 (2.7)	4.19 (1.28–13.72)	0.018
Atrial dilatation (*N* = 1138)				
no	890 (78.2)	6 (0.7)	Reference	
yes	248 (21.8)	3 (1.2)	1.77 (0.44–7.09)	0.417

*N* = number, HF = heart failure, HR = hazard ratio, CI = confidence intervals, MRI = magnetic resonance imaging; LIC = liver iron concentration.

**Table 3 diagnostics-13-00890-t003:** Comparison of the three groups identified on the basis of the number of CMR abnormalities.

Variable	No CMR Abnormalities (*N* = 488)	One to Three CMR Abnormalities (*N* = 617)	Four CMR Abnormalities (*N* = 7)	*p*-Value
Women, *N* (%)	263 (53.9%)	308 (49.9%)	0 (0.0%)	0.010
Age (years)	30.56 ± 8.68	31.49 ± 8.24	32.90 ± 6.28	0.315
Age at start of regular transfusions (years)	2.12 ± 2.84	2.43 ± 4.84	2.50 ± 1.91	0.280
Chelation starting age (years)	4.43 ± 3.96	4.81 ± 4.14	4.00 ± 2.12	0.313
Pre-transfusion hemoglobin (g/dL)	9.58 ± 0.58	9.61 ± 0.77	9.27 ± 0.52	0.335
Serum ferritin (ng/mL)	1283.28 ± 1148.91	1681.25 ± 1669.89	1359.50 ± 945.66	0.001
Global heart T2* (ms)	35.17 ± 6.68	24.49 ± 12.89	10.01 ± 2.37	<0.0001
Number of segments with T2* < 20 ms	1.17 ± 1.99	7.13 ± 6.90	15.29 ± 0.76	<0.0001
MR LIC (mg/g/dw)	6.86 ± 7.49	10.54 ± 13.14	9.51 ± 7.69	<0.0001
LV EF (%)	64.87 ± 5.13	59.11 ± 7.18	40.16 ± 12.32	<0.0001
RV EF (%)	64.83 ± 5.73	58.66 ± 8.40	40.73 ± 13.21	<0.0001
LV EDVI (mL/m^2^)	84.16 ± 15.98	89.80 ± 20.32	136.14 ± 23.09	<0.0001
RV EDVI (mL/m^2^)	80.95 ± 15.38	85.71 ± 20.21	130.39 ± 16.57	<0.0001
LV cardiac index (L/min/m^2^)	3.83 ± 1.05	3.59 ± 1.08	3.72 ± 0.41	<0.0001
Replacement myocardial fibrosis, *N* (%)	0 (0.0)	180 (29.2)	7 (100.0)	<0.0001
Left atrial area (cm^2^/m^2^)	12.57 ± 2.37	13.26 ± 2.76	15.52 ± 2.48	<0.0001
Right atrial area (cm^2^/m^2^)	11.73 ± 2.05	12.34 ± 2.53	13.97 ± 2.45	0.001

*N* = number, MRI = magnetic resonance imaging, LIC = liver iron concentration, LV = left ventricular, EF = ejection fraction; RV = right ventricular, EDVI = end-diastolic volume index.

## Data Availability

The data underlying this article cannot be shared publicly due to privacy reasons. The data will be shared on reasonable request to the corresponding author.

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
