# Peer review of "Multi-Parametric Cardiac Magnetic Resonance for Prediction of Heart Failure Death in Thalassemia Major"

_diagnostics, 2023, doi:10.3390/diagnostics13050890_

Round 1

Reviewer 1 Report

This is very well and clearly written artciel about Multi-parametric Cardiac Magnetic Resonance as a predictor of heart failure death in thalassemia major. The study was performed on large group of patients, results are clear, I can only suggest in dsicussion add more data about iron toxicity (cardio, and cells toxicity) according to new references 2022-2023

I recommend to accept this article.

Author Response

This is very well and clearly written artciel about Multi-parametric Cardiac Magnetic Resonance as a predictor of heart failure death in thalassemia major. The study was performed on large group of patients, results are clear, I can only suggest in dsicussion add more data about iron toxicity (cardio, and cells toxicity) according to new references 2022-2023

I recommend to accept this article.

A: We thank the Reviewer for the positive feedback.

More data about iron toxicity have now been added in the Discussion as follows.

“Iron excess can be detrimental to human cells through the production of hydroxyl rad-icals via the Haber–Weiss–Fenton reactions, that cause oxidative damage to cellular components like lipids, proteins, and DNA (ref). Free iron can directly interact and interfere with a variety of ion channels of cardiomyocytes in-cluding the L-type calcium channel, the ryanodine-sensitive calcium channel, volt-age-gated sodium channel, and delayed rectifier potassium channel, making cardio-myocytes particularly vulnerable to iron overload. Excessive production of reactive oxygen species can also directly induce ferroptosis (a non-apoptotic mode of cell death) in cardiomyocytes by catalysing the oxidation of phospholipids in the cell membrane (ref).

Reviewer 2 Report

The topic of the manuscript is very intriguing. However, the authors should address the following major points:

- The definition of the diagnosis of advanced heart failure should be better clarified. The authors report criteria which are not in line with current ESC recommendations. Moreover, it is not clear if all the criteria reported had to be satisfied before the enrollment. If this is the case, the mean LVEF (37%) is greater than the cut-off of 30%. Moreover, only 76% of patients were taking loop diuretics. This is not in line with a selection of advanced patients or it indicates a suboptimal therapy.

- Clinical presentation of patients should be reported (acute decompensated chronic heart failure, right heart failure or pulmonary edema).

- The diuretics use should be better detailed. This is very relevant. The dose as well as the use of different class of diuretics (only loop diuretics, loop diuretics + thiazides, loop diuretics + acetazolamide) should be reported. 

-  No data are provided about clinical congestion and/or hypoperfusion. The relationship between 2h UNa and congestion should be considered. Was it related with a more rapid resolution of congestion? Moreover, data about patients at discharge should be reported. In particular, the percentage of patients still presenting clinical signs of congestion. Once again the relationship with UNa should be analyzed.

- Although it is a sample of patients affected by an advanced heart failure characterized by a high rate of events, its number is very small as stated by the authors in the limitation section. 

Author Response

We would like to thank the Reviewer for the encouraging feedback and constructive critique and for the effort regarding this manuscript. We have addressed each of the concerns raised by the Reviewer, which have substantially improved the manuscript.

The topic of the manuscript is very intriguing. However, the authors should address the following major points:

- The definition of the diagnosis of advanced heart failure should be better clarified. The authors report criteria which are not in line with current ESC recommendations. Moreover, it is not clear if all the criteria reported had to be satisfied before the enrollment. If this is the case, the mean LVEF (37%) is greater than the cut-off of 30%. Moreover, only 76% of patients were taking loop diuretics. This is not in line with a selection of advanced patients or it indicates a suboptimal therapy.

A: Current ESC recommendations have now been taken into account. As stated in 2.1 subsection, 5.9% of patients were retrospectively excluded from the analysis due to a history of heart failure. The presence of an LV dysfunction only did not represent an exclusion criterion.  

We have some difficulty in fully addressing the Reviewer issue “the mean LVEF (37%) is greater than the cut-off of 30%”, as baseline LVEF values reported in our study in all patient classes are different (>  48% in all classes)”. We would kindly ask the reviewer to clarify the point to allow as to better reply.

We have also some difficulty in fully understanding the ”Moreover, only 76% of patients were taking loop diuretics.” Reviewer issue, since no data about the use of diuretics were present in the manuscript. Generally, diuretics are used cautiously and only in the late stages of disease in thalassemia patients, as suggested in guidelines (Cappellini 2008, https://pubmed.ncbi.nlm.nih.gov/24308075/). Hence, the recording of diuretics use was not mandatory in the study and, unfortunately, representative data are not available.

- Clinical presentation of patients should be reported (acute decompensated chronic heart failure, right heart failure or pulmonary edema).

A: More details have been added in the text as follows. “Ten patients had HF with reduced EF at echocardiography. The majority of them presented to the healthcare provider with a reduction in their effort tolerance due to dyspnea and/or fatigue. One patient presented not only with fatigue but also with chest pain and tachycardia and had elevated troponin level. Two patients presented with palpitations. Two patients had a chronic heart failure, diagnosed >1 year after the CMR scan and rapidly worsened. One patient had an HF with mildly reduced EF. One patient had HF with preserved EF and had evidence of structural heart disease.”.

- The diuretics use should be better detailed. This is very relevant. The dose as well as the use of different class of diuretics (only loop diuretics, loop diuretics + thiazides, loop diuretics + acetazolamide) should be reported. 

A: As previously stated, unfortunately we don’t have significant data about the use of the diuretics in our TM population as they are not commonly used in this kind of population.  Therefore, we cannot specify the class of diuretics.

-  No data are provided about clinical congestion and/or hypoperfusion. The relationship between 2h UNa and congestion should be considered. Was it related with a more rapid resolution of congestion? Moreover, data about patients at discharge should be reported. In particular, the percentage of patients still presenting clinical signs of congestion. Once again the relationship with UNa should be analyzed.

A: Unfortunately, no data about the 2h UNa is available in our thalassemic patients.

- Although it is a sample of patients affected by an advanced heart failure characterized by a high rate of events, its number is very small as stated by the authors in the limitation section. 

A: The number of events was really small since we focused only on deaths for heart failure. This limitation is acknowledged in the limitation section “The small number of HF deaths that occurred during the follow-up ….”

Round 2

Reviewer 2 Report

I would like to apologize with the authors, for an inexplicable mistake in some points of my review. I've no further comments.